# Missense variant analysis in the TRPV1 ARD reveals the unexpected functional significance of a methionine

Grace C. Wulffraat[1], Sanjana Mamathasateesh[1], Rose Hudson[1], Benjamin He[1], Andrés Jara-Oseguera[2], Eric N. Senning[1]*

**1** Department of Neuroscience, University of Texas at Austin, Austin, Texas, United States of America,
**2** Department of Molecular Biosciences, University of Texas at Austin, Austin, Texas, United States of America

* esen@utexas.edu

## Abstract

The Transient Receptor Potential Vanilloid sub-type 1 (TRPV1) is an ion channel that is activated by heat, extracellular protons, oxidation, and it is implicated in various aspects of inflammatory pain. In this study, we uncover that residue M308 in the TRPV1 ankyrin repeat domain (ARD) stands out from most other buried ARD residues because of the greater number of human missense variants at this position while maintaining a high degree of conservation across species and TRPV channel subtypes. We use mutagenesis and electrophysiology to examine this apparent discrepancy and show that substitutions at position M308 that preserve or reduce side-chain volume have no effect on channel function, whereas substitutions with larger or more polar residues increase channel activity in response to capsaicin or temperature. Substitution of M308 with a histidine bestows channels with pH-dependence that is different from wild type, consistent with the side-chain at position 308 exerting an influence on channel gating. We speculate that M308 is highly conserved because its side-chain could serve as a target for oxidation-dependent modification. On the other hand, we show that a previously described splice variant of TRPV1 that relies on M308 as a start codon diminishes surface expression of co-transfected full-length TRPV1 in HEK293 cells. Together, our findings reveal a functionally important conserved site within the ARD of TRPV1 that could have roles in oxidation-dependent channel regulation as well as tuning the number of active channels in the membrane by enabling expression of a shorter dominant-negative splice variant.

## Introduction

The Transient Receptor Potential subfamily V member 1 (TRPV1) cation channel is a $Ca^{2+}$ permeable ion channel expressed in neurons of the peripheral nervous system, where it has a central role in nociception and inflammatory pain [1]. Multiple

**Data availability statement:** All relevant data are within the manuscript and its Supporting Information files.

**Funding:** This work was supported by start-up funds from the University of Texas at Austin (E.N.S.) and the NSF-MCB grant no. 2129209 (E.N.S.).

**Competing interests:** The authors have declared that no competing interests exist.

components of the "inflammatory soup," a term coined to denote a mixture of neu-rotrophins (i.e., NGF or bradykinin), cellular metabolites and low pH that permeate injured tissue, directly or indirectly upregulate TRPV1 activity and expression and are therefore a direct cause of inflammatory and chronic pain [2–5].

Whereas ion conduction and the binding of vanilloids and many other important ligands occur within the transmembrane domain, the role of the large cytoplasmic N-terminal ankyrin repeat domain in channel function is unclear. TRPV1 is sensitive to its chemical environment and exhibits positive modulation by a range of oxidizing reagents, and functional studies have implicated several cysteines located in the cytosolic N-terminal region of the channel known as the ankyrin repeat domain (ARD) in mechanisms of oxidative channel regulation [6,7]. The ARD also mediates channel desensitization via interaction with $Ca^2+$-Calmodulin, as well as channel sensitization by ATP [8,9]. Structural studies of TRPV1 provide a framework for understanding how these distal sites in cytosolic domains would orchestrate conformational changes that influence channel activity [10]. These structural data, together with computational studies, have even implicated the ARD in the mechanism by which TRPV channels sense temperature [11–14]. However, a rigorous mechanistic understanding of how the dynamics at the ARD lead to a change in TRPV1 gating is absent.

Sequence comparison across protein orthologs is an important tool that can narrow down structural regions that could be implicated in channel function [15]. We have taken advantage of human population genomic data to generate structure-function hypotheses regarding TRPV1 and have previously described that the number of distinct amino acid variants observed in the human population at each TRPV1 N-terminal ankyrin repeat domain position is positively correlated with their solvent accessible surface area (SASA) [16]. Our prior study led us to hypothesize that sites with strong deviations from the correlation are likely to have functional relevance. By mining TRPV1 human genomic variant data in the Genome Aggregation Database (gnomAD), we have identified buried amino acid clusters within the ARD that have an unusually high number of amino acid variants given their low solvent accessible surface area (SASA), suggesting that these regions may be of functional importance. One particular fold between ankyrin repeats 4–5 of the ARD drew our immediate attention because of a partially buried methionine (M308) and alanine (A256) that exhibited unusual variant properties despite being highly conserved in TRPV1 across species. Side chains from both residues extend into a buried pocket and are within 5 Å of each other in a closed channel structure configuration (**Fig 1**). Moreover, numerous amino acids with side chains buried in the ankyrin repeat 4–5 fold have greater than expected numbers of missense variant amino acid types within the human population. Here we examine the mutational landscape of methionine M308 in this fold and attribute functional changes caused by some residue substitutions to the constraints imposed by the confinement of the side chains at this position. We show that substitution of M308 to histidine modified the influence of intracellular protons on the channel as compared to WT. Regulation of TRPV1 through oxidation of the M308 position could easily incur a similar rearrangement of side chain packing in the fold between ankyrin repeats 4–5 that is seen with protonation of a histidine at

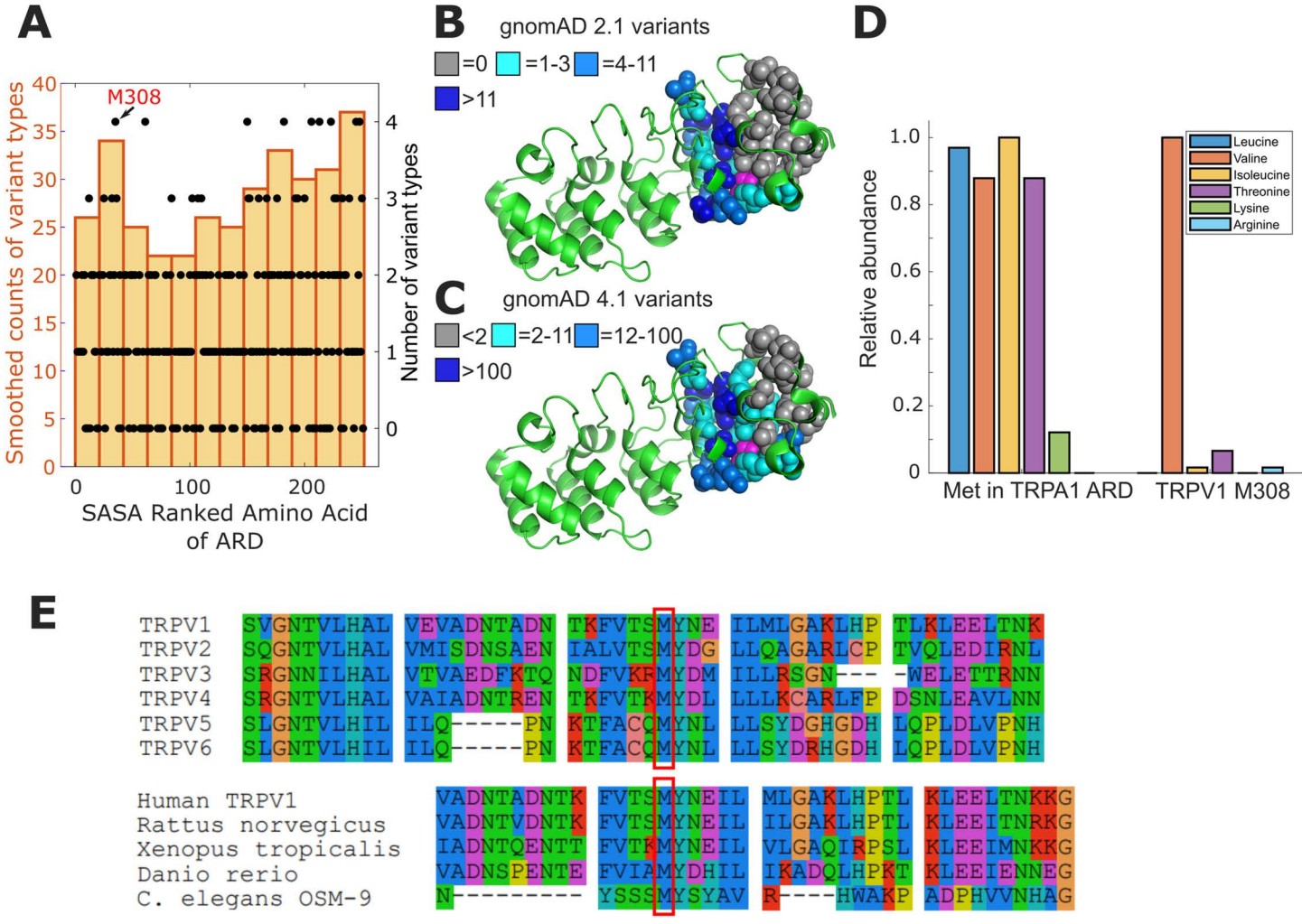

**Fig 1. High density of missense variants between ankyrin repeats 4 and 5 of TRPV1. (A)** Bar plot illustrates the smoothed counts of variant types (20 positions wide) for SASA ranked amino acid in the ARD of rat TRPV1 (residues 111-361 based on pdb file: 7LP9). The number of different types of amino acids arising from missense mutations at a given position are shown as filled circle markers (black) with corresponding y-axis on the right side. Residue M308 (numbering in rat TRPV1) is the upper most point (x = 35) on the left with 4 types of variants (see S1 Fig). **(B)** Structural model of the TRPV1 ARD (PDB: 7LP9) and buried side chains of interest shown as space filling spheres with coloring to indicate absolute number of missense variants in gnomAD 2.1. M308 has 4 alternate types of missense variants and is shown in magenta. Side chains of interest in between ankyrin repeats 4,5, and 6 are colored according to the legend. **(C)** Identical structural model as shown in B with side-chains colored according to missense variant counts in gnomAD 4.1. **(D)** Relative abundance of missense variants in gnomAD for all methionine (Met) positions in ARD (1-530) of TRPA1 and at M308 of TRPV1. The most common amino acid encoded by missense variants in the 16 Met residues of the ARD in human TRPA1 is isoleucine (total of 33). According to gnomAD 4.1, the most common missense variant of M308 in human TRPV1 is valine, with 121 alleles. Counts were normalized to most common missense amino acid. **(E)** Sequence alignment of human TRPV subfamily (upper set TRPV1-6) and TRPV1 orthologues (lower set) in local proximity to M308, outlined with a red box (rat TRPV1 numbering). Alignment sequence IDs: *H. sapiens* TRPV1, NP_542435.2; *H. sapiens* TRPV2, NP_057197.2 *H. sapiens* TRPV3, NP_001245134.1; *H. sapiens* TRPV4: XP_054228773.1; *H. sapiens* TRPV5, NP_062815.3; *H. sapiens* TRPV6, NP_061116.5; *R. norvegicus* TRPV1, O35433.1; *X. tropicalis* TRPV1, NP_001243521.1; *D. rerio* TRPV1, NP_001119871.1; *C. elegans* OSM-9, CCD61335.1.

this position or with substitutions to side-chains having a larger size. Alternatively, we also find that having a methionine at position 308 could be important to function as an alternative start codon, giving rise to a shorter dominant negative splice variant that would affect the number of functional TRPV1 channels in the plasma membrane of sensory neurons. [17,18]

## Materials and methods

### Missense variant analysis

Data were obtained from the gnomad database (https://gnomad.broadinstitute.org/) as csv files and processed by in-house MATLAB (Mathworks, MA) programs that are freely available through github (https://github.com/esenlab/variant-analysis) and described in our previous study [16]. Structure files were rendered for publication with PyMOL (Schrödinger, Germany).

### Cell culture

HEK293T/17 (ATCC: CRL-11268) cells were incubated in Dulbecco's Modified Eagle Medium, supplemented with 10% fetal bovine serum, 50 µg streptomycin, and 50 units/mL penicillin, at 37°C and 5% CO2. Cells were passaged onto poly-L-lysine treated 25 mm coverslips. Cells were allowed at least 2h to settle onto the slips before being transfected using Lipofectamine 2000 (Life Technologies) as described in the manufacturer's instructions. Cells were transfected with pcDNA3.1 containing TRPV1 wild type and mutants. Following transfection, $Ca^2+$ imaging experiments were carried out after 24h, and electrophysiology experiments were completed 24-48h after transfection. For electrophysiology experiments co-transfection of the TRPV1 construct of interest was done alongside a GFP expressing plasmid (JSM-164 NW10 3C-GFP-HS).

### Molecular biology

In vivo assembly (IVA) methods of site-directed mutagenesis were done as previously described [16]. In the case of M308I, the construct was obtained from Azenta Life Sciences (South Plainfield, NJ) with our TRPV1.pcDNA3 provided as starting material. For all in-house mutagenesis, our wild-type TRPV1.pcDNA3 and TRPV1-C157A.pcDNA3 vector (Gift from A. Jara-Oseguera) was used as a template with (IVA) overlapping primers that contain the desired mutation (see primers for mutagenesis spreadsheet). The full sequence of each completed construct was confirmed using Sanger sequencing.

### Electrophysiology

Currents were recorded at room temperature in symmetric divalent-free solutions (130 mM NaCl, 3 mM HEPES, 0.2 mM EDTA, pH 7.2) using fire polished borosilicate glass pipettes with filament (outer diameter 1.5 mm, inner diameter 0.86 mm; Sutter, MA). Pipettes were heat polished to a resistance of 3.5–7 MΩ using a micro forge. Cells were transfected with channel constructs and replated after 18–24 hours on 12 mm coverslips and placed in divalent-free solution in the chamber. 1 µM and 5 µM capsaicin solutions were prepared from a 5.3 mM capsaicin stock in ethanol. The same stock was diluted in the recording solution to 0.5 mM to make 0.03 µM, 0.1 µM, and 0.3 µM capsaicin solutions. Excised inside-out patches were positioned at the opening of a tube in a "sewer pipe" configuration with 8 tubes connected to a rod controlled by an RSC-200 rapid solution changer (BioLogic, France). Each perfusion barrel was briefly primed by applying pressure through the syringe reservoir before the next barrel was positioned in front of a patch. Capsaicin solutions (0.03, 0.1, 0.3, 1, 5 and 10 µM) were perfused for 40-120s each. For some experiments, the same solutions were perfused into the chamber using open, gravity-driven reservoirs for 100-120s each. TRPV1-M308H pH dependent experiments were conducted using divalent-free HEPES buffer as described above with pH adjusted as indicated in experiments (nominal pH: 7.2; high pH: 8.0; low pH: 6.5). Currents were measured with an Axopatch-200A e-phys amplifier at sweep intervals of 1 second stepping the voltage from resting potential (0mV) to −80 mV (400 ms), then up to +80 mV (400 ms) and back to resting potential.

Currents were filtered at 5 kHz and recorded at 20 kHz using an Axopatch 200A amplifier (Axon Instruments, Inc.) and PatchMaster software (HEKA). Data was analyzed in PatchMaster and IGOR (Wavemetrics, OR). Individual patch-clamp recordings with a complete set of stable current measurements in all capsaicin solutions were normalized to the

maximal current with 5 µM capsaicin. Normalized data was fitted to the Hill equation to arrive at the $EC_{50}$ parameter: $I = I_{max}([ligand]^n/(EC_{50}^n + [ligand]^n))$, where n was permitted to float. Statistical significance was determined by two-tailed Student's t-test. For pH dependent measurements in macroscopic TRPV1 wild type excised patches the approximate open probability was calculated by normalizing the current under each pH condition with the maximal response measured with 5 µM capsaicin. Statistical evaluation of the change in currents due to shifting pH conditions was done with a one-sample t-test applied to the difference in currents for individual patches.

For single channel data sets, the +80 mV segment of the sweep was concatenated in MATLAB (Mathworks, MA). The current baseline was subtracted off using the xarray-graph plug-in with Matlab (https://github.com/marcel-goldschen-ohm/xarray-graph) and channel open probability was determined by partitioning the distribution of events by a cutoff that was half the current for the single channel open state. A notebook in Python was implemented for this purpose. In some instances, the MATLAB plug-in smBEVO was used to correct single channel current drift and evaluate state changes in the current profile. This plug-in is available from the Goldschen-Ohm lab through github (https://github.com/marcel-goldschen-ohm/smBEVO). As described, in the accompanying readme.txt file of the smBEVO plug-in, we optimized the filtering parameters of smBEVO by visual assessment with each data set until the modeled fit approximated behavior for the observed number of single channels.

Temperature-activation experiments were carried out in the whole-cell configuration of the patch clamp using a custom-modified deltaT temperature-control system (Bioptechs, Butler PA). The bottom of the recording chamber in this system is coated with an electrically conductive material that allows to pass current through to cause rapid and homogeneous changes in temperature on the opposite side of the chamber where the cells are located. Our system was modified to achieve more rapid temperature-changes. Temperature can also be cooled below ambient by flowing ice water through a closed metal loop that fits within the chamber. We recorded the temperature close to the pipette tip using a small thermistor bead. For these experiments we used pipettes with resistances between 0.5–1 MΩ and the same solutions as for inside-out recordings. After achieving the whole cell configuration, we lifted cells up and started flowing water through the cooling ring to drop the temperature to 10–15°C. Temperature was then increased up to 60°C and currents were recorded until the patch broke. Currents were recorded in response to pulses to ±100 mV with a duration of 400 ms from a holding potential of 0 mV. Pulses were applied with a frequency of 1 Hz. Data was acquired at 5 kHz and low pass filtered at 1 kHz, using a dPatch amplifier (Sutter, Novato CA). The steady-state current magnitude at the end of each pulse to +100 mV was used to quantify current-temperature relations for each experiment.

## Calcium imaging

Cells were grown on 25-mm coverslips and then incubated for 30 min at room temperature with Fluo-4 (AM; Thermofisher) at a concentration of 3 µM. The cells were then rinsed with Hepes buffered Ringer's (HBR) solution (in mM, 140 NaCl, 4 KCl, 1.5 MgCl2, 5 D-glucose, 10 HEPES, and 1.8 CaCl2 and pH adjusted to 7.4 with NaOH) and allowed to rest for another 30 min in HBR at room temperature. The cells were imaged on a Nikon Eclipse Ti microscope using a 10×objective. For each slip, a brightfield image, a fluo-4 image, and a 3-min fluo-4 fluorescence movie with exposures of 100-ms and 0.5-s intervals were captured. During the time sequence, HBR is initially perfused throughout the chamber. Perfusion is switched to 500 nM capsaicin in HBR at 30 s, and at the 2 min mark, 3 µM ionomycin is added to the chamber to equilibrate calcium between inside and outside the cell in order to obtain the maximal Fluo-4 response. The HBR and 500 nM capsaicin in HBR were perfused into the chamber using open, gravity-driven reservoirs, and 500 uL of 3 µM ionomycin was Pipetted into the chamber via micro-Pipette. The data obtained during these experiments were analyzed using Image J and Matlab (Mathworks, MA).

## Cell response counts in TRPV1 truncation constructs

Truncated constructs (TRPV1-preDelM308 and TRPV1–5'sv) were co-expressed with TRPV1 wild type constructs as 25% to 75% plasmid DNA weight ratio: TRPV1 wild type (25%) and PLCδ1-PH-SNAPtag (75% DNA; a plasmid

membrane marker for PIP2); TRPV1 wild type (25%) and TRPV1-preDelM308 (75%); TRPV1 wild type (25%) and TRPV1–5'sv (75%). Before counting, the brightness and contrast were adjusted using FIJI's automatic adjustment. Contrast and brightness adjustment was made with the "auto" feature in contrast and brightness control window at the point in the movie after the capsaicin response was induced, but before administration of ionomycin (around frame 200). This was done to make cells visible and distinguishable. A screenshot was then taken of the region with this brightness/contrast setting. Cells were marked with a pink line on this screenshot as they were counted to avoid any repeat counts or missed cells. Cell brightness was characterized from the beginning of the movie to the end of the movie (after ionomycin addition) for this region. If any cell did not increase in brightness due to either capsaicin or ionomycin addition (this includes cells which began and remained bright), it was crossed out in white and subtracted from the total cell count. Lastly, the cells with positive capsaicin responses were identified. This was done by comparing the brightness of cells in the first frame of the movie and the frame used to establish brightness/contrast settings. Cells with a qualitative change in brightness between these two points were marked as capsaicin responses. The fraction of cells responding to capsaicin was determined by dividing the number of capsaicin-responding cells by the number of total healthy (ionomycin or capsaicin-responding) cells. This process was repeated by two different people to check for consistency in the counting process. To ensure that data was not affected by variability across transfection days, we normalized the ratio of responses in experimental sets (TRPV1 WT with TRPV1-preDelM308 and TRPV1 WT with TRPV1–5'sv) by same day controls (TRPV1 WT with PLCδ1-PH-SNAPtag). Therefore, plotted values in **Fig 4D** are compared to a maximal TRPV1/PLCδ1-PHh response of 1.0.

### Live cell staining of TRPV1 surface expression

HEK293T cells were transfected with 50:50 DNA ratios of TRPV1 WT:TRPV1-exCellHalo in a control sample and TRPV1WT:TRPV1Δ1–308-exCellHalo in a test sample. TRPV1Δ1–308-exCellHalo was also transfected in isolation. Cells were stained after 24 hours with the Promega dyes Alexa488-HaloTag and JF549HaloTag, which are respectively cell impermeable and cell permeable. Coverslips were incubated in complete media with 1 μM of each dye type for 30 minutes at 37 degrees C before the media was exchanged and the cells recovered at room temperature for 10 minutes, followed by immediate imaging.

## Results

### A valine at position 308 in TRPV1 is a frequent missense allele in the human population

Our previous work highlighted a positive correlation between solvent accessible surface area (SASA) of the amino acids and missense variants within the ankyrin repeat domain (ARD) that are present in the human population [16]. When ranked by their fractional SASA scores, the amino acids of the ARD show an increasing trend in the types of amino acids coded by missense variant sequences found in the human population (yellow bar plot, **Fig 1A** and **S1 Fig**). This variant analysis of TRPV1 revealed a region between ankyrin repeats 4 and 5, containing a methionine (M308) and an alanine (A256), that is crowded with variant amino acids (see colored side chains of **Fig 1B**) and contrasts sharply with the reduced number of variants observed in the neighboring fold between ankyrin repeats 5–6 (see grey side chains of **Fig 1B**). M308 is represented by a black marker (**Fig 1A** indicated with arrow), and the four variant types associated with that position set it apart as an outlier (missense variants from gnomAD release 4.1.0; Val:121, Thr:8, Ile:2, Arg:2). Nevertheless, we found that M308 is highly conserved in TRPV1 orthologs and across the TRPV sub-family (**Fig 1E**). The observed missense variants for M308 in the human population appear to be sharply defined by the size of the side chain (Met: 105.1 ml/mole) [19]. A slightly larger residue (107.5 ml/mole) such as isoleucine, is far less common than the valine (91.3 ml/mole) (i.e., there are 121 valine alleles and only 2 isoleucine). To investigate this in greater detail, we compared the missense allele counts for the M308 position in TRPV1 against all 16 methionines present in the ankyrin repeat domain of TRPA1 (**Fig 1C**). The missense variants in the TRPA1 methionines distribute across hydrophobic amino

acids relatively evenly, suggesting that the prevalence of valine for TRPV1 M308 is functionally meaningful. We therefore hypothesized that a functional effect is observed when M308 is exchanged for amino acids larger than valine.

## Capsaicin sensitivity of M308 mutants points to side chain constraints in TRPV1 ARD

To test our theory that substitution of the M308 fold is functionally tolerated up to a specified side-chain volume, we replaced the methionine with Ala, Val, Ile, and Trp and experimentally determined the capsaicin dose-response relation, and the associated $EC_{50}$, for each mutant (Fig 2A,B).Mutating M308 to tryptophan resulted in a non-functional channel.

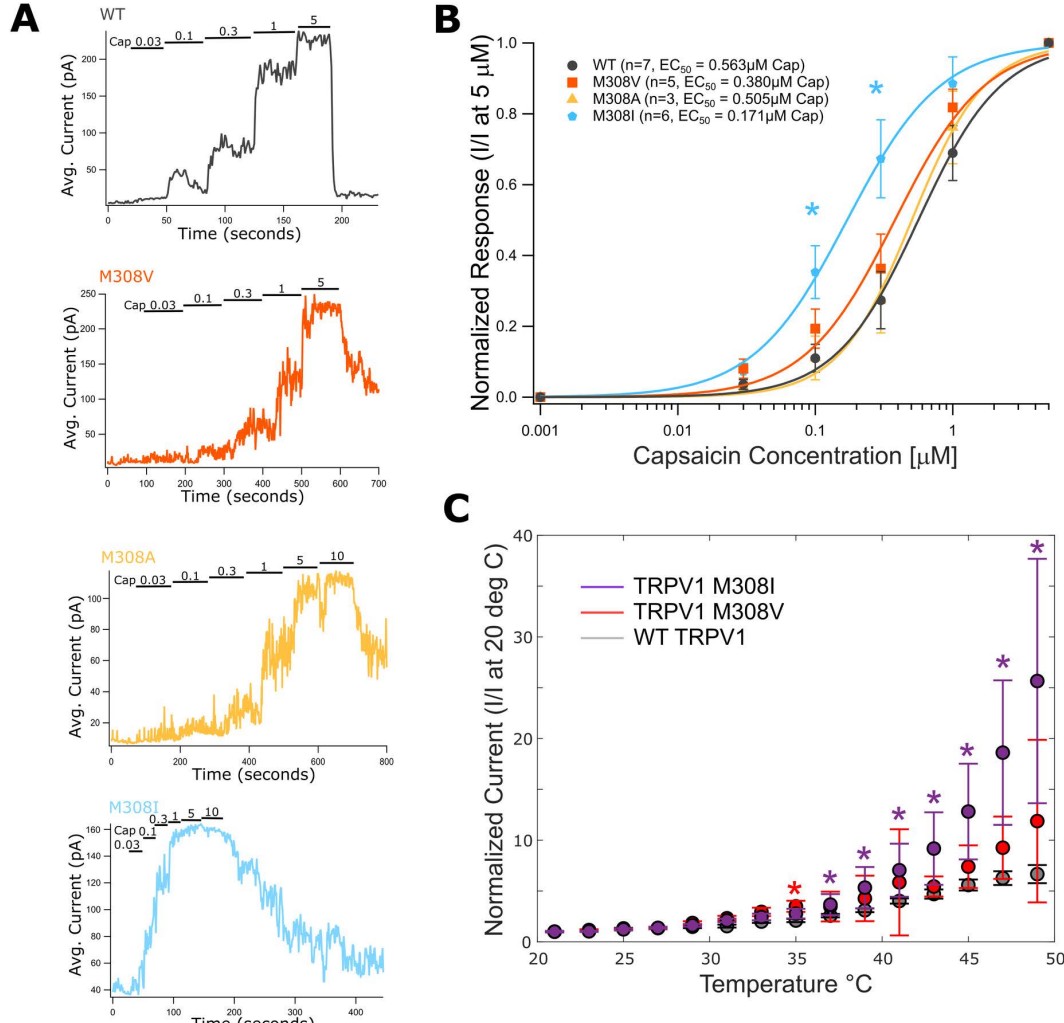

**Fig 2. Functional characterization of TRPV1-M308 with larger and smaller side chain substitutions by capsaicin dose-response and temperature activation experiments. (A)** Representative traces of capsaicin dose response relations in TRPV1-M308 mutants. Solution exchanges with different concentrations of capsaicin given in µM above traces. **(B)** Summary data of capsaicin dose response experiments shown as log-linear plots and fitted with a Hill function to acquire $EC_{50}$ values. The TRPV1-M308I mutant average responses is significantly different from wild type TRPV1 at two capsaicin concentrations, 0.1 and 0.3 µM (p = 0.0117, 0.0124). **(C)** Whole-cell current-temperature relations at +100 mV showing median temperature dependent activation of TRPV1-M308I (purple), TRPV1-M308V (red) and wild-type TRPV1 (grey) expressed in HEK293T/17 cells. Individual traces were normalized to the mean steady-state current magnitude measured at room temperature. Data points are shown at 2°C intervals and error bars are **S.**E.M. (TRPV1 WT, n = 11; TRPV1-M308V, n = 11; TRPV1-M308I, n = 5). Significant differences between wild type and TRPV1-M308I are indicated at 37, 39, 41, 43, 45, 47, 49 and 45°C (*; p = 0.033, 0.018, 0.009, 0.006, 0.003, 0.003, 0.006) and TRPV1-M308V at 35°C (*; p = 0.026. Student's t-test).

Alanine and valine substitutions could not be distinguished from wild type in their capsaicin sensitivity measured by patch-clamp while the isoleucine substitution at M308 incurred a significant increase in sensitivity to capsaicin. Therefore, our findings suggest that the fold between ankyrin repeats 4–5 imposes a unique space constraint on the side chain at position 308. Channels with an Ala, Val, or Met side-chain at position 308 behave just like WT, but insertion of hydrophobic residues Ile and Trp that are larger than methionine [19] have a strong effect on TRPV1 channel function, respectively increasing the channel's sensitivity to stimuli or completely compromising function.

### Temperature dependent activation of M308 mutations is consistent with their capsaicin sensitivity

We tested the temperature activation profiles of TRPV1 wild type, TRPV1-M308V and TRPV1-M308I to examine whether the isoleucine substitution also increased sensitivity to heat. Although there was considerable variability in the data, we observed a clear trend where M308I currents increased more steeply than those of WT or M308V channels (Fig 2C). These observations provide further support to the hypothesis that there are steric constraints for the side-chain at position 308, and that even a modest increase in side-chain volume has the functional effect of increasing channel activity.

### Protonation of M308H modulates TRPV1 function

We observed that substitutions at M308 with side chains smaller than methionine retained WT channel function, whereas substitution with the slightly larger, and branched isoleucine increased channel activity in response to capsaicin, which leads us to hypothesize that covalent modification of the M308 side-chain could result in channel activation or sensitization. We first tested whether covalent modification of a cysteine introduced at position 308 in the C157A background construct of TRPV1, which is insensitive to MTS reagent modification [6], had any effect on channel function. Although the TRPV1-C157A/M308C construct expressed as a functional channel with a dose response profile to 0.3 and 5 μM capsaicin that was indistinguishable from the C157A background, we did not detect any effects on the currents measured upon exposure to MTSEA (S2 Fig) or MTSET (S3 Fig). This observation suggests that M308C either reacts with the reagents but this does not alter channel function or the site is not accessible to these reagents, the latter being consistent with the low SASA for this position measured from structures. We reasoned that whereas inaccessible to large cysteine-reactive molecules, the side-chain at position 308 might be accessible to protons, and its protonation might be sufficient to induce a change in channel activity. We therefore proceeded to mutate the M308 to a histidine, which is protonatable across a physiological pH range. It was previously shown that in addition to a strong sensitivity to extracellular protons [20], the TRPV1 channel exhibits increased activity with elevated intracellular pH [21]. Consistent with these findings, we show that WT TRPV1 expressed in HEK293T inside-out patches and activated with sub-saturating capsaicin has slightly reduced or unchanged current at pH 6.5 but increased currents at pH 8 (Fig 3A). The TRPV1-M308H channel is functional, but it suffers from inconsistent and poor expression in HEK293T cells. We did not detect any grossly altered capsaicin sensitivity for the M308H mutant evaluated at pH 7.2 (S4 Fig). Therefore, nearly all measurements of channel activity for this mutant are single-channel recordings (expanded views of the single channel behavior [i.,ii.,iii.] are shown below the full extent of the recording in Fig 3B). In contrast with WT channels, we observed that the open probability of M308H channels increased at pH 6.5, which promotes histidine protonation, and decreased at pH 8, which promotes deprotonation (Fig 3B,C,D). Although most of the data collected for M308H was of single channels, we collected one complete recording of a multi-channel patch expressing M308H, which had a pH-dependent response profile consistent with what we observed in single channel patches (S5 Fig). This trend was apparent in all patches, even though some single M308H channels exhibited increased activity at all pH values relative to WT channels. Overall, these findings suggest that protonation/deprotonation of the histidine side chain at position 308H results in detectable changes in channel activity that are absent in WT channels. These observations establish that steric or electrostatic perturbations at position M308 may influence channel activity, but that a methionine is not required for normal channel function, raising the question of why M308 is so highly conserved.

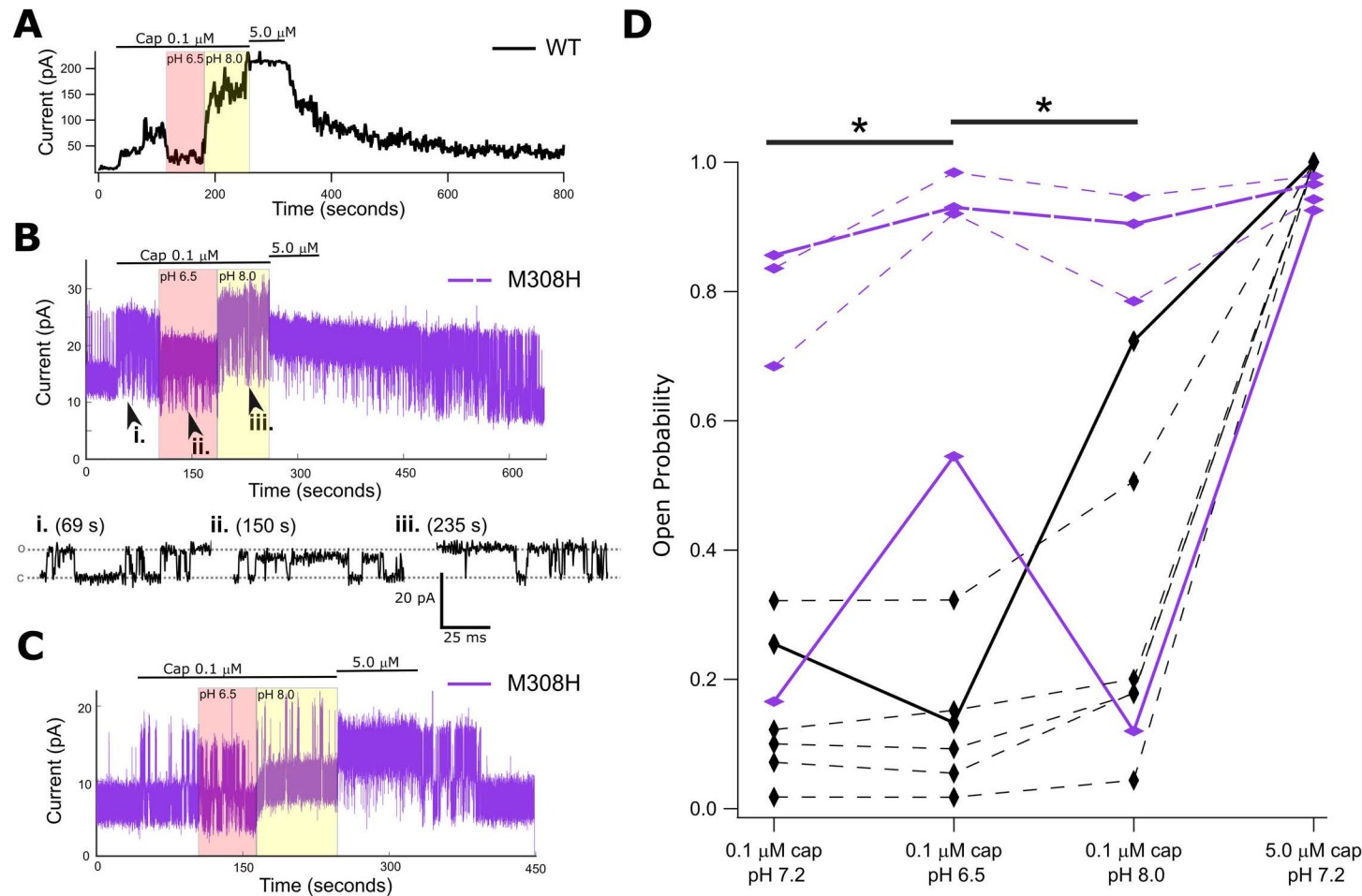

**Fig 3. TRPV1-M308H modulation by intracellular pH. (A)** Wild type TRPV1 macro patch recording shows a large change in current response with intracellular solution exchange to low (6.5) or high pH (8.0) relative to nominal pH (7.2) while capsaicin (Cap) is maintained at 0.1 µM. **(B)** Single channel recording of TRPV1-M308H in excised patch showing changes in current with simultaneous switching to different capsaicin concentration or pH of solution applied to inside out patch. Zoomed in regions of the current record show single channel opening and closing events (i.-iii.). There is a notable reduction in channel unitary conductance at lower pH and leak current is also noticeably increased at pH 8.0. **(C)** Single channel recording of TRPV1-M308H with greatest observed effect on $P_o$ upon changing pH of solution applied to inside out patch. **(D)** Summary data of TRPV1 wild type (black) and M308H (purple) currents in response to 0.1 µM capsaicin and varying inner leaflet pH. Vertical diamond markers indicate macro patches and horizontal diamond markers indicate single channel recordings. Open probability calculated as normalized value to maximal current response with 5 µM capsaicin in TRPV1 wild type, macro patches and as fraction of current data points in "open" state relative to all data points for each condition in single channel, M308H patches – see methods. Thick lines are color coordinated with individual traces shown in A-C. A significant difference between TRPV1 wild type and M308H current responses is observed in repeated measures when switching 0.1 µM capsaicin containing solutions to pH 6.5 from pH 7.2 or from pH 6.5 to pH 8.0 (p = 0.095 in both with Mann-Whitney U-test).

### TRPV1 splice variant with start codon at M308 has a dominant-negative effect on full-length TRPV1 activity

The existence of mRNA for a splice variant of TRPV1 with a reading frame starting at position M308 has been described and found to have a dominant-negative effect on the magnitude of currents measured from cells where this shorter variant is co-expressed with WT. [18] However, it is unclear if the effect is due to a reduction in channel expression or channel function. A physiologically relevant role for this shorter splice variant could provide an explanation for why the methionine at position 308 is so highly conserved across species even though other amino acids at this position (i.e., Ala, Val,

Cys – see S3 Fig) result in a channel with no or subtle changes to its function under the experimental conditions that we explored. To address this question, we first assessed whether the shorter isoform of TRPV1 gave rise to functional channels using $Ca^{2+}$-imaging (Fig 4B). We focused on two different constructs, one that exactly recapitulates the splice variant that was previously identified (TRPV1–5'sv, Fig 4A and C, structure with missing segments shown in purple), and another where only the first 307 residues of the N-terminus are deleted, which we term TRPV1-preDelM308 (Fig 4A). We conducted baseline $Ca^{2+}$ imaging experiments with only the truncated constructs to confirm previously published results that these constructs are non-functional. [18] To test the dominant negative potential of each truncated construct, we counted the number of capsaicin responding HEK293T cells that co-express the indicated combination of TRPV1 wild type and TRPV1-preDelM308 or TRPV1–5'sv relative to a global calcium response from ionomycin and calculated response ratios to TRPV1 wild-type co-expressed with a bystander construct (pcDNA3.1 with PLCδ1-PH-SNAPtag) to balance DNA amounts in the transfection step. We found that the addition of TRPV1–5'sv to TRPV1 wild type in the transfection step significantly reduced the number of cells that responded to capsaicin (Fig 4D). The TRPV1-preDelM308 isoform added to TRPV1 wild type also had a reduced number of cells responding to capsaicin compared to WT alone but did not surpass significance at the $p = 0.05$ level at the given DNA ratio.

**TRPV1 splice variant with start codon at M308 is not detected on cell surface**

From our previous experiment that examined the activity of TRPV1 co-expressed with shortened isoforms, the question remained whether a reduction in TRPV1 activity was caused by a loss in surface expression or if a heteromeric channel was expressed with altered function. We next determined the cellular expression pattern of a shortened isoform in the absence and presence of co-expressed full-length TRPV1. To accomplish this we utilized a fluorescent reporter construct for TRPV1 surface expression in which a self-labeling HaloTag domain was fused into the extracellular face of the pore of the channel, so that surface-expressed channels can be specifically labeled using a HALO-reactive membrane-impermeable fluorescent probe [22]. A labeling strategy with cell permeable and impermeable HaloTag ligands could therefore distinguish between channels reaching the cell surface and channels retained in intracellular compartments. This labeling strategy with HaloTag ligands showed that full-length TRPV1-exCellHalo is present at the plasma membrane and in cellular compartments, as would be expected for the cellular expression trafficking of the full-length channel (**Fig 4E**). By co-expressing our HALO-tag enabled shortened TRPV1 isoform (TRPV1Δ1–308-exCellHalo) with and without TRPV1 wild type we could discern if the truncated construct's dominant negative effect arises through a heteromeric complex at the plasma membrane or if it is linked to upstream steps in channel expression. The TRPV1Δ1–308-exCellHalo accumulates in cell compartments but does not accumulate at the cell surface even when co-expressed with TRPV1 wild-type construct (**Fig 4E**). Together with the TRPV1 $Ca^{2+}$ imaging data to establish the dominant-negative effect of isoform co-expression on activity, the extracellular label findings implicate a mechanism by which the shorter isoforms suppress the cell surface expression of full-length TRPV1 homomeric channel complex.

## Discussion

Our recent study that related human missense variants with structural data in TRPV1 prompted further investigations of how solvent accessible surface area (SASA) correlated with missense variant frequency. Although SASA measurements trended with a greater frequency of missense mutations at a specific amino acid position, there were outliers to this paradigm, including the methionine in position 308 of the rat TRPV1 sequence (S1 Fig). Interestingly, the methionine is also highly conserved across orthologues, which contrasts sharply with its associated number of variants observed in the population. Based on these data, we developed an early hypothesis about the functional role of M308: 1) The fold between ankyrin repeats 4 and 5 of the TRPV1 ARD can accommodate a side chain that closely matches the volume and hydrophobicity of methionine and 2) disruption of the packing in this fold through post-translational modification of the methionine would modulate channel activity.

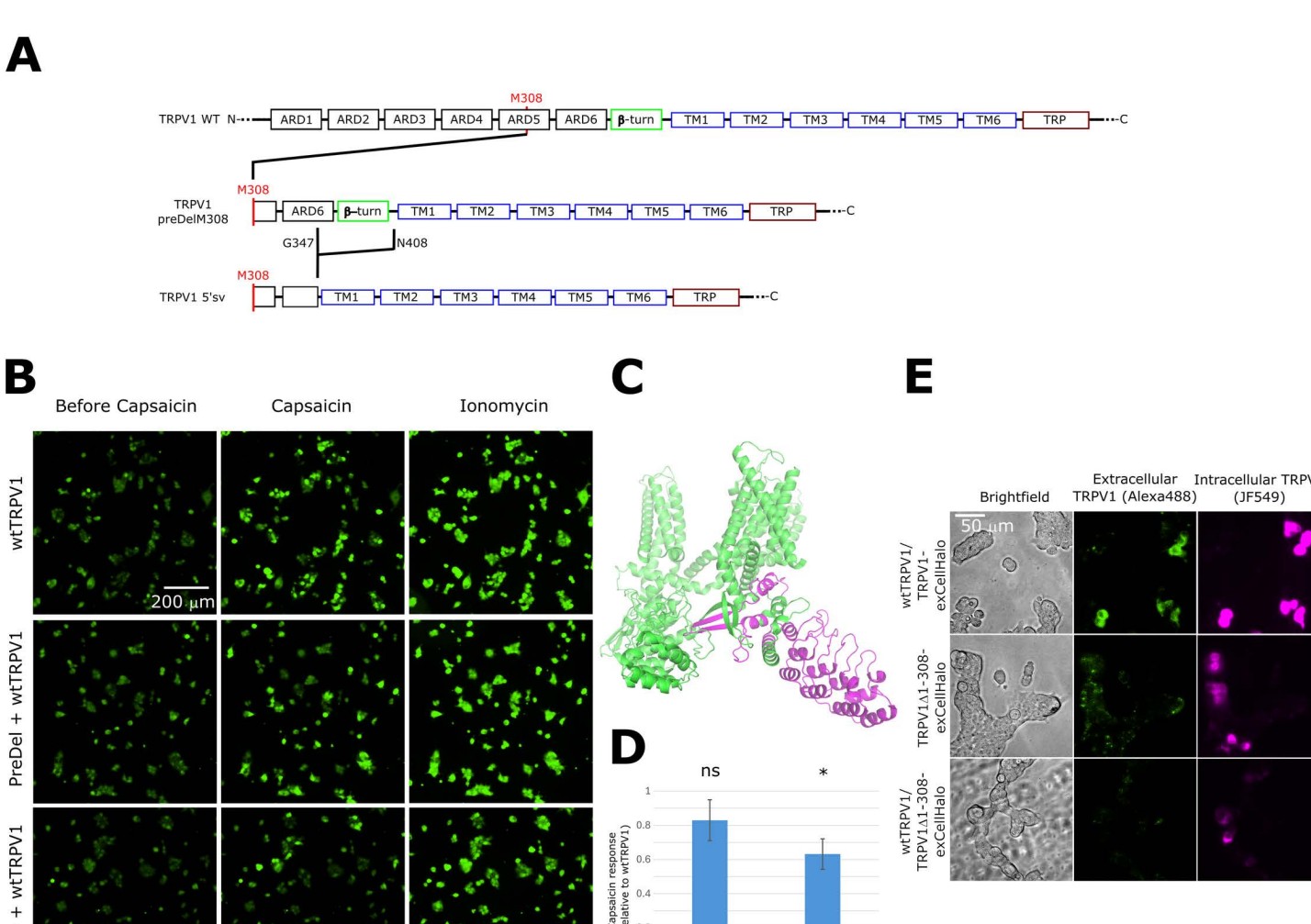

**Fig 4. Ca²⁺ imaging of TRPV1 wild type co-expressed with TRPV1-preDelM308 or TRPV1-5'sv. (A)** Three TRPV1 constructs used in HEK293T co-transfection experiments to establish dominant negative expression effect of truncated isoforms. Residue numbering is maintained from rat TRPV1. TRPV1 preDelM308 isoform is an N-terminal truncated construct that relies on M308 as the first amino acid. TRPV1 5'sv also begins at M308 and has an internal deletion between residues G347 and N408. **(B)** Representative images from fluorescence video series of HEK293T/17 cells co-expressing: TRPV1 wild type with a vector expressing PLCδ1-PH-SNAPtag; TRPV1 wild type with TRPV1-preDelM308 (Δ1-307): or TRPV1 wild type with TRPV1-5'sv. Addition of 0.5 μM capsaicin activates channels, resulting in Ca²⁺ influx and an increase in the fluorescence intensity of a calcium reporter. Ionomycin (5 μM) equilibrates internal, cellular Ca²⁺ concentration with that of the extracellular buffer. Total DNA in cotransfections is contributed as 50% from each indicated construct by weight. **(C)** Structural model of TRPV1 (PDB: 7LP9), where purple regions highlight either deletions of both 1-307 (majority of TRPV1 ARD) and 348-407 (beta turn) in TRPV1-5'sv or only deletion of 1-307 in TRPV1-preDelM308. **(D)** Summary of cell counting analysis from representative data in panel B and larger sample size across several experiments. Ratio of cells responding to capsaicin relative to ionomycin in each movie are normalized to the results in same-day experiments of TRPV1 wild type co-transfected with PLCδ1-PH-SNAPtag. Statistical significance was observed between TRPV1 wild type and TRPV1-5'sv (*; one-sample Student's t test: p-value = 0.0037) but not with TRPV1-preDelM308 (ns; one-sample Student's t test: p-value = 0.0652). **(E)** Live cell imaging of extracellular HaloTag TRPV1 constructs expressed in HEK293T cells and labeled with HaloTag fluorophores. Experiments with two constructs are co-transfected with 50% of each plasmid DNA by weight. Labeling with cell permeable (JF549) and impermeable (AlexaFluor-488) HaloTag ligands shows whether expressed channels reside exclusively on intracellular membranes or also on the plasma membrane.

Substituting M308 with valine or alanine, which occupy a slightly smaller volume, did not alter the capsaicin or temperature sensitivity of the channel. However, exchanging the methionine for the slightly larger isoleucine altered capsaicin and temperature activation profiles of the channel. We reasoned that the sensitivity to the size of side chain at position 308 is narrowly defined to match methionine to fulfill a role in post-translational modification. Although our initial attempts to test this with a M308C substitution and modification by MTS reagents were rebuffed, we proceeded to test the hypothesis that protonation of the partially buried side-chain in M308H would affect channel gating. A recent study of histidine orientation in a buried fold showed that the packing of the side chain is strongly dependent on its protonation state [23]. We therefore surmised that the sensitivity to intracellular pH in the mutant could indeed be linked to the protonation of the histidine side chain at the 308 position and a perturbation to the fold between ankyrin repeats 4 and 5. The ARD is structurally adjacent to the C-terminal domain of a neighboring subunit in TRPV1, and we speculate that conformational changes induced by the expansion of the fold around M308 would propagate across the inter-subunit interface with downstream effects on channel gating (**Fig 5A**). Our previous studies on the interface between the ARD and C-terminal domain have shown that weakening the contacts between these two domains shifts the equilibrium of the channels to favor opening. Because our current study implicates expansion of the pocket around M308 in channel activation, we speculate that that pocket expansion could weaken the interface between N- and C-termini from neighboring subunits. We find further support for the pocket-expansion hypothesis when examining the distribution of missense variants across ankyrin repeats 4–6. There are numerous missense variants that can be assigned to side-chains between ankyrin repeats 4 and 5, whereas the number of missense variants between ankyrin repeats 5 and 6 is sharply reduced (**Fig 1B**). If the increased number of missense variant positions between ankyrin repeats 4 and 5 can be interpreted as flexibility, then we speculate that this region may have a hinge-like property. Post-translational modification of the methionine, in this case, would expand the pocket to flex the hinge while the neighboring ankyrin repeats 5 and 6 retain a fixed assembly to magnify the change in the preceding ARD fold 4–5. From a structural perspective, it is not surprising to find that apo structures of TRPV1 maintain a well resolved ARD that extends outward from the pore beyond the 5th ankyrin repeat when temperatures are below the activation threshold [11,24]. However, at higher temperatures (48°C) the EM map in the region of ankyrin repeats 1–4 is lost (PDB: 7LPC; See Extended **Figure 4** of Kwon et al.) [12]. Similarly to the TRPV1 structure obtained at high temperature, a structure obtained at nominal temperature (PDB: 7L2N) in the presence of the highly potent agonist resiniferatoxin (RTx) also has missing EM density for ankyrin repeats 1–4. If ankyrin repeats 4 and 5 demarcate a flexible boundary, this would translate to increased ARD movements that would not be resolvable in the electron density maps.

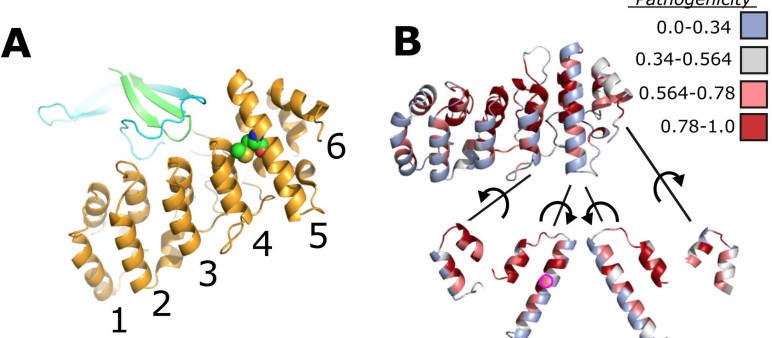

**Fig 5. Stability of TRPV1 intersubunit interactions between ARD and C-terminal region. (A)** Structural view of the ARD (yellow) interface with the neighboring subunit's beta turn (green) and the C-terminal domain (cyan). The ankyrin repeats are numbered along the bottom of the ARD. **(B)** Alphamissense pathogenicity coloring scheme overlaid onto ARD. Intrasubunit interfaces between ankyrin repeats 4-5 and 5-6 shown with connecting lines and exposed by rotation of the ankyrin repeat according to spiral arrow. Pathogenicity score values imposed on structural representation of TRPV1 (Uniprot identifier: Q8NER1) obtained from the Hegelab website and described by Tordai and coworkers [25]. The color table for pathogenicity is rated as follows: blue (0-0.34), likely benign; grey (0.34-0.564), ambiguous; light red (0.564-0.78), likely pathogenic; dark red (0.78-1.0), likely pathogenic. The M308 position is indicated with a magenta circle.

Genetic sequence data remains an incredible resource to the ion channel biophysics community [15,26,27]. With the more recent availability of human population exome and genome data, there has been an intense effort to mine these for their medical insight and potential determination of disease pathogenicity [28–30]. We along with others have developed tools which examine the structural associations between variant positions in proteins that allow for structure-function predictions, potentially allowing us to test hypotheses about protein function [31,32].

Alphamissense is perhaps the most ambitious project to use amino acid sequence data in a survey of the pathogenicity attributable to each amino acid in the human proteome, and its availability as a community resource allowed us to access the pathogenicity score of all M308 substitutions [33]. The experimental e-phys validation of the Alphamissense predictions provides a critical assessment of such computational tools. In the Alphamissisense data set hosted by the Hegelab, the human TRPV1 M309 position lists all amino acid substitutions except valine as likely pathogenic (S1 Table). Within the scope of M308 mutations that we studied, isoleucine, histidine, and tryptophan had significant changes from TRPV1 wild type function, which matched the Alphamissense designation of these substitutions as "likely pathogenic" (Fig 5B). The methionine to valine change had only a mild functional effect on temperature dependent activity and falls in line with the predicted "likely benign" classification. We were surprised to see that the change to alanine was predicted "likely pathogenic" although there was nothing remarkably different from TRPV1 wild type in its functional characterization. We reasoned that testing channel activity with capsaicin alone was not sufficient to characterize TRPV1 functional properties pertaining to pathogenicity, or the pathogenicity prediction by Alphamissense was inaccurate. Turning to our experiments that explore the dominant negative potential of shortened TRPV1 constructs revealed a significant drop in cellular responses to capsaicin when wild type TRPV1 is contransfected with the 5'sv construct. If we were to believe that the methionine is an important start codon for expression of the truncated TRPV1–5'sv construct as demonstrated in experiments shown in Fig 4, then all the substitutions for methionine would have to be pathogenic for this mechanism to have a physiologically relevant regulatory role. Although there exists a precedent in the literature for the shorter isoform, indisputable evidence in the existing exomic or genomic sequence data to support this is lacking. Nevertheless, the methionine is remarkably salient across TRPV1 orthologues and other members of the TRPV sub-family, which beckons the question of why Alphamissense would label the valine missense variant as likely benign. Based on our experimental results, the methionine at residue 308 is optimally positioned to undergo post-translational modification that may induce a TRPV1 conformational change and modulate activity. In addition, translation of a shorter TRPV1–5'sv isoform beginning at M308 alters full-length TRPV1 channel expression in the plasma membrane. Therefore, two different, established properties of a methionine could play different roles in TRPV1 regulation.

## Supporting information

**S1 Fig. Variant number versus ranked SASA score.** Fractional SASA score (pdb structure file: 7lp9) is plotted against SASA ranked amino acids of the ARD (residues 111–361) in rat TRPV1 (purple). Missense variant type count in gnomAD 4.1 for each position (black marker) is indicated on the right-side y-axis. SASA ranked amino acid X = 35 corresponds to M308 in the rat TRPV1 sequence and as indicated on the top left box. The Matlab script for generating this plot is described in Mott et al. (2023) and available through GitHub (*esenlab*/variant-analysis).
(PDF)

**S2 Fig. Application of MTSEA to TRPV1-M308C/C157A excised patches does not affect channel activity. (A)** Macroscopic current voltage clamp experiment with application of capsaicin and MTSEA as indicated over traces. Control TRPV1-C157A with 1 mM MTSEA and co-application of 0.3 µM capsaicin. **(B)** TRPV1-C157A/M308C with 1 mM MTSEA and co-application of 0.3 µM capsaicin.
(PDF)

**S3 Fig. Application of MTSET to TRPV1-M308C/C157A-expressing excised patches does not affect channel activity. (A)** Macroscopic current voltage clamp experiment with application of capsaicin and MTSET as indicated over current traces. Control TRPV1-C157A with 100 µM MTSET and co-application of 0.3 µM capsaicin. **(B)** Control TRPV1-C157A with 1 mM MTSET and co-application of 0.3 µM capsaicin. **(C)** TRPV1-C157A/M308C with 100 µM MTSET and co-application of 0.3 µM capsaicin. **(D)** TRPV1-C157A/M308C with 1 mM MTSET and co-application of 0.3 µM capsaicin. (PDF)

**S4 Fig. TRPV1M308H dose response data with single channel and multi-channel patches. (A)** Currents collected in voltage sweeps with voltage set to +80mV. Because the voltage sweeps include a -80mV and +80mV segment, we used a macro to extract the +80mV segments and concatenate these in the presented and analyzed data (described in the Methods section). Current traces with single and double channel contributions are colored, and the multi-channel patch is black. Notable properties of the green trace is high activity for two channels in the wash buffer before and after capsaicin application. **(B)** Summary plot of the channel activity illustrated in data traces shown in (A). The multi-channel patch is plotted with a line and single channel data is plotted with colored markers. The purple trace shows two active channels until the application of the 5 µM solution when one channel becomes silent. The purple 5µM concentration data point in (B) is based on a single channel remaining active in the purple trace. (PDF)

**S5 Fig. Modulation of currents by pH on inner leaflet of excised membrane patch from cells expressing reduced wild type TRPV1 and TRPV1-M308H. (A)** Inside out patch with 5 WT channels from an experiment also included in wild type sample from Fig 3D. (B) Inside out patch with 5 TRPV1-M308H channels with similar current response profile to single channel experiments shown in Figure 3. (C) Summary data of normalized responses in experiments from panels A and B. (PDF)

**S1 Table. Alphamissense pathogenicity of M309 position in TRPV1.** Excerpt of Alphamissense scores assigned to M309 in human TRPV1 (M308 in the rat TRPV1 sequence). SNV, indicates with "y" whether a single nucleotide variant is known to exist. (PDF)

**S1 Data. A zip file containing the data which was used to analyze and present the results shown in figures of the manuscript.** (ZIP)

**S2 Data. Supplemental Table of Primers: Table of DNA oligos that were used as primers for generating the mutants presented in this study.** (XLSX)

## Acknowledgments

We thank Dr. Sharona Gordon for TRPV1.pcDNA3.1 and reading an early version of the manuscript and Dr. Jason McLellan for JSM-164 NW10 3C-GFP-HS (GFP). We also thank Dr. Shai Silberberg for a conversation that started the ball rolling on the mysterious M308 residue of TRPV1.

## Author contributions

**Conceptualization:** Andrés Jara-Oseguera, Eric Senning.

**Data curation:** Grace C. Wulffraat, Sanjana Mamathasateesh, Rose Hudson, Benjamin He, Andrés Jara-Oseguera.

**Formal analysis:** Grace C. Wulffraat, Rose Hudson, Andrés Jara-Oseguera, Eric Senning.

**Funding acquisition:** Eric Senning.

**Investigation:** Grace C. Wulffraat, Eric Senning.

**Methodology:** Grace C. Wulffraat, Eric Senning.

**Project administration:** Eric Senning.

**Resources:** Eric Senning.

**Software:** Grace C. Wulffraat, Sanjana Mamathasateesh, Eric Senning.

**Supervision:** Andrés Jara-Oseguera, Eric Senning.

**Validation:** Grace C. Wulffraat, Rose Hudson, Benjamin He, Andrés Jara-Oseguera.

**Visualization:** Grace C. Wulffraat, Sanjana Mamathasateesh, Rose Hudson, Eric Senning.

**Writing – original draft:** Grace C. Wulffraat, Eric Senning.

**Writing – review & editing:** Grace C. Wulffraat, Rose Hudson, Andrés Jara-Oseguera.

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
