## [Decision Letter · Decision Letter 0]

30 Apr 2025

PONE-D-25-14924Missense variant analysis in the TRPV1 ARD reveals the unexpected functional significance of a methionine.PLOS ONE

Dear Dr. Senning,

Thank you for submitting your manuscript to PLOS ONE. The manuscript has been evaluated by two experts in the field. Both reviewers indicated that the manuscript reports interesting data. However, the reviewers also identified several problems. Thus, although the manuscript has merit, it does not fully meet PLOS ONE’s publication criteria as it currently stands. Therefore, we invite you to submit a revised version of the manuscript that addresses the concerns of the reviewers.

We look forward to receiving your revised manuscript.

Kind regards,

Alexander G Obukhov, Ph.D.

Academic Editor

PLOS ONE

Journal Requirements:

We thank Dr. Sharona Gordon for TRPV1.pcDNA3.1 and Dr. Jason Mclellan for JSM-164 NW10 3C-GFP-HS (GFP). We also thank Dr. Shai Silberberg for a conversation that started the ball rolling on the mysterious M308 residue of TRPV1. This work was supported by start-up funds from the University of Texas at Austin (E.N.S.) and the NSF-MCB grant no. 2129209 (E.N.S.).

This work was supported by start-up funds from the University of Texas at Austin (E.N.S.) and the NSF-MCB grant no. 2129209 (E.N.S.).

5. We noted in your submission details that a portion of your manuscript may have been presented or published elsewhere. A previous version of this manuscript was uploaded to the Bioarxiv server. Please clarify whether this [conference proceeding or publication] was peer-reviewed and formally published. If this work was previously peer-reviewed and published, in the cover letter please provide the reason that this work does not constitute dual publication and should be included in the current manuscript.

6. We note that you have indicated that there are restrictions to data sharing for this study. PLOS only allows data to be available upon request if there are legal or ethical restrictions on sharing data publicly. For more information on unacceptable data access restrictions, please see http://journals.plos.org/plosone/s/data-availability#loc-unacceptable-data-access-restrictions.

7. In the online submission form, you indicated that our data include larger sets of images acquired in calcium imaging movies. We therefore will provide reasonable access to these for a specific range of images upon request.

8. Please amend your list of authors on the manuscript to ensure that each author is linked to an affiliation. Authors’ affiliations should reflect the institution where the work was done (if authors moved subsequently, you can also list the new affiliation stating “current affiliation:….” as necessary).

9. We note that you have included the phrase “data not shown” in your manuscript. Unfortunately, this does not meet our data sharing requirements. PLOS does not permit references to inaccessible data. We require that authors provide all relevant data within the paper, Supporting Information files, or in an acceptable, public repository. Please add a citation to support this phrase or upload the data that corresponds with these findings to a stable repository (such as Figshare or Dryad) and provide and URLs, DOIs, or accession numbers that may be used to access these data. Or, if the data are not a core part of the research being presented in your study, we ask that you remove the phrase that refers to these data.

10.Please include captions for your Supporting Information files at the end of your manuscript, and update any in-text citations to match accordingly. Please see our Supporting Information guidelines for more information: http://journals.plos.org/plosone/s/supporting-information.

Reviewers' comments:

Reviewer's Responses to Questions

**Comments to the Author**

1. Is the manuscript technically sound, and do the data support the conclusions?

Reviewer #1: Yes

Reviewer #2: Partly

2. Has the statistical analysis been performed appropriately and rigorously? 

Reviewer #1: Yes

Reviewer #2: Yes

3. Have the authors made all data underlying the findings in their manuscript fully available?

Reviewer #1: No

Reviewer #2: Yes

4. Is the manuscript presented in an intelligible fashion and written in standard English?

Reviewer #1: Yes

Reviewer #2: Yes

5. Review Comments to the Author

Reviewer #1: The general quality of the data presented in the manuscript is good, and the study addresses a relevant scientific question. However, the presentation of the data lacks clarity and, at times, appears sloppy. This significantly affects the readability and interpretation of the results. I have several major comments that need to be addressed to improve the manuscript.

Please see the attached review for more details.

Reviewer #2: In the manuscript “Missense variant analysis in the TRPV1 ARD reveals the unexpected functional significance of a methionine.”, by Grace C. Wulffraat, Sanjana Mamathasateesh, Rose Hudson1, Benjamin He, Andrés Jara-Oseguera1, Eric N. Senning. Authors provide new data about isolates residue M308 in the ankyrin repeat domain of TRPV1 channel, which exhibits a large number of missense variants in humans while being conserved across species. Mutagenesis experiments have shown that substitutions in M308 can affect channel function, with larger or more polar residues increasing activity. In addition, the conservation of M308 may be related to its role in redox-dependent modifications, and a splice variant lacking M308 reduces TRPV1 expression on the surface of HEK293 cells. These data suggest that M308 is critical for channel regulation and expression dynamics.

Although the study is of undoubted interest and expands our knowledge about functions of TRPV receptors to increase the level of validity of the material and publication in the journal, it needs some improvements.

In the Materials and Methods section, the missense variant analysis, then electrophysiology. I would put the information about the object of study (Cell Culture) at the beginning. The electrophysiology section goes on to cell culture and it's not clear what kind of culture it is.

All the Materials and Methods sections are quite detailed, but the molecular biology section contains mostly references to literature, it should be expanded to include information on the experimental approaches used in the study.

Capsaicin was diluted in ethanol. Potentially ethanol can have an effect on cells . It is necessary to specify what concentration of solvent was used and whether the same concentration of solvent was added to the control solutions. If not, explain why this can be neglected.

In the experiments with temperature change a modified setup was used, it is necessary to indicate how fast and with what accuracy it was possible to change the temperature in the experiment.

The Materials and Methods section lacks information about the method of staining of the preparation in Fig. 4 E and the microscopic technique with which it was obtained. There is no scale in Fig. 4 E.

There are two large sections in the Materials and Methods are devoted сalcium imaging experiments. But in the Results and Discussion section I found very little information regarding the description of these experiments. It is necessary to clearly describe why they were performed and what the authors wanted to show. Why ionomycin was used, what it allows to reveal, what follows from the obtained data. I did not find in the discussion section any specific conclusions on this block of results.

The same applies to data obtained by electrophysiological methods, and there is no discussion of these data or conclusions from these experiments in the discussion section.

I think it is also good to add to the discussion information about the physiological significance of the obtained results

With all the suggested additions, the paper can certainly be accepted for publication.

6. PLOS authors have the option to publish the peer review history of their article (what does this mean? ). If published, this will include your full peer review and any attached files.

**Do you want your identity to be public for this peer review?** For information about this choice, including consent withdrawal, please see our Privacy Policy .

Reviewer #1: No

Reviewer #2: No

---

## [Author Response · Author response to Decision Letter 1]

15 Jul 2025

Please see the attached "response to reviewers" document

---

## [Decision Letter · Decision Letter 1]

13 Aug 2025

Missense variant analysis in the TRPV1 ARD reveals the unexpected functional

significance of a methionine.

PONE-D-25-14924R1

Dear Dr. Senning,

We’re pleased to inform you that your manuscript has been judged scientifically suitable for publication and will be formally accepted for publication once it meets all outstanding technical requirements.

Kind regards,

Alexander G Obukhov, Ph.D.

Academic Editor

PLOS ONE

Reviewers' comments:

Reviewer's Responses to Questions

**Comments to the Author**

1. If the authors have adequately addressed your comments raised in a previous round of review and you feel that this manuscript is now acceptable for publication, you may indicate that here to bypass the “Comments to the Author” section, enter your conflict of interest statement in the “Confidential to Editor” section, and submit your "Accept" recommendation.

Reviewer #2: All comments have been addressed

2. Is the manuscript technically sound, and do the data support the conclusions?

Reviewer #2: Yes

3. Has the statistical analysis been performed appropriately and rigorously? 

Reviewer #2: Yes

4. Have the authors made all data underlying the findings in their manuscript fully available?

Reviewer #2: Yes

5. Is the manuscript presented in an intelligible fashion and written in standard English?

Reviewer #2: Yes

6. Review Comments to the Author

Reviewer #2: In general, the author tried to take into account all the comments of the reviewers. The manuscript can be accepted for publication

7. PLOS authors have the option to publish the peer review history of their article (what does this mean? ). If published, this will include your full peer review and any attached files.

**Do you want your identity to be public for this peer review?** For information about this choice, including consent withdrawal, please see our Privacy Policy .

Reviewer #2: No

---

## [Editor Report · Acceptance letter]

PONE-D-25-14924R1

PLOS ONE

Dear Dr. Senning,

I'm pleased to inform you that your manuscript has been deemed suitable for publication in PLOS ONE. Congratulations! Your manuscript is now being handed over to our production team.

Kind regards,

on behalf of

Dr. Alexander G Obukhov

Academic Editor

PLOS ONE